# Dual-initiating and living frustrated Lewis pairs: expeditious synthesis of biobased thermoplastic elastomers

Yun Bai[1], Huaiyu Wang[1], Jianghua He[1], Yuetao Zhang [1✉] & Eugene Y.-X. Chen [2]

Biobased poly(γ-methyl-α-methylene-γ-butyrolactone) (PMMBL), an acrylic polymer bearing a cyclic lactone ring, has attracted increasing interest because it not only is biorenewable but also exhibits superior properties to petroleum-based linear analog poly(methyl methacrylate) (PMMA). However, such property enhancement has been limited to resistance to heat and solvent, and mechanically both types of polymers are equally brittle. Here we report the expeditious synthesis of well-defined PMMBL-based ABA tri-block copolymers (tri-BCPs)—enabled by dual-initiating and living frustrated Lewis pairs (FLPs)—which are thermoplastic elastomers showing much superior mechanical properties, especially at high working temperatures (80–130 °C), to those of PMMA-based tri-BCPs. The FLPs consist of a bulky organoaluminum Lewis acid and a series of newly designed bis(imino)phosphine superbases bridged by an alkyl linker, which promote living polymerization of MMBL. Uniquely, such bisphosphine superbases initiate the chain growth from both P-sites concurrently, enabling the accelerated synthesis of tri-BCPs in a one-pot, two-step procedure. The results from mechanistic studies, including the single crystal structure of the dually initiated active species, detailed polymerizations, and kinetic studies confirm the livingness of the polymerization and support the proposed polymerization mechanism featuring the dual initiation and subsequent chain growth from both P-sites of the superbase di-initiator.

[1] State Key Laboratory of Supramolecular Structure and Materials, College of Chemistry, Jilin University, Changchun, Jilin, China. [2] Department of Chemistry, Colorado State University, Fort Collins, CO, USA. ✉email: ytzhang2009@jlu.edu.cn

Exploring sustainable alternatives to the petroleum-based polymers has attracted intense attention due to the concerns about accelerated depletion of finite nature resources[1–5]. However, to realize the practical application of sustainable polymers, it is insufficient to just synthesize polymers from biorenewable monomers, the biobased polymer should exhibit superior or performance-advantaged properties to those of the petroleum-based polymers for real-world applications[4,6]. As the cyclic analog of methyl methacrylate (MMA), renewable methylene butyrolactone-based vinylidene monomers, such as α-methylene-γ-butyrolactone (MBL) and γ-methyl-α-methylene-γ-butyrolactone (MMBL), are of particular interest in exploring the prospects of substituting the petroleum-based methacrylate monomers for sustainable polymer production[7–10]. Compared to PMMA, these renewable methylene butyrolactone-based polymers typically exhibited increased resistance to common organic solvents, heat, and scratch[11–13]. In particular, their glass-transition temperature ($T_g$) values are much higher than that of PMMA: 195 °C for poly(MBL) (PMBL); 227 °C for PMMBL, 105 °C for PMMA[14–18]. Despite these enhanced resistance to heat and solvent, much like PMMA, P(M)MBL materials are mechanically brittle, which largely limited their practical applications. It is well known that ABA type tri-block copolymers (tri-BCPs) composed of hard end block A and soft midblock B can self-assemble into thermoplastic elastomers (TPEs) with a wide range of applications by combining thermoplastics' processability with elastomers' ductility and toughness[19–21]. Hence, we envisioned that renewable PMMBL could be employed as the high $T_g$ hard segments for TPEs, which would not only solve the brittleness issue, but also produce high-temperature TPEs by taking advantage of its high $T_g$. However, it remains as a challenge to synthesize PMMBL-based TPEs containing a soft methacrylate block, because after the completion of MMBL polymerization, the growing chain end is not nucleophilic enough to initiate the polymerization of less reactive methacrylate monomers[22–24]. Although PMBL-based TPEs are known, they were synthesized by atom transfer radical polymerization (ATRP) involving presynthesis and isolation of Br-(soft-block) polymer-Br macroinitiators for subsequent ATRP of MBL[25–27]. So far, we are unaware of reports on PMMBL-based TPEs achieved by living polymerization strategy.

Owing to their unique, unquenched reactivities, frustrated Lewis pairs (FLPs) not only demonstrated well-established utilities in small molecular chemistry[28–36] but also exhibited unique applications in polymer synthesis[37–47]. By selecting a suitable organoaluminum Lewis acid (LA) with appropriate steric hindrance and acidity, we achieved the first example of living polymerization of methacrylates by an authentic FLP[48]. The field of Lewis pair (LP) polymerization has grown considerably over the past decade[49–51]. Recently, by fine-tuning of both the steric and electronic properties of the organophosphorus superbase as the paring Lewis base (LB) in the FLP, we successfully synthesized ultra-high molecular weight PMMA with number-average molecular weight ($M_n$) up to 1927 kg/mol at room temperature (RT)[52] and a series of sequence-controlled multiblock copolymers in 30 min at RT, including a tripentacontablock copolymers ($n = 53$, $k = 4$, $dp_n = 50$) with the record number of blocks to date[53]. It is known that, in a living polymerization system, the polymer architectures can be controlled by the structure of the initiator[54–57]. Owing to the facile structural tunability of these organophosphorus superbases[58,59], we hypothesized that specifically designed bisphosphine superbases could form a FLP with a suitable LA, which would potentially serve as a dual-initiating and living LP polymerization system. Achieving this goal would solve the current challenge facing the synthesis of PMMBL-based TPEs by turning the traditional three sequential-monomer-addition steps of the tri-BCP synthesis into an accelerated, one-pot, two-step procedure. Success in developing this new method would then create PMMBL-based TPEs to explore performance-advantaged properties, especially mechanical toughness much desired for acrylics, for such biobased polymers.

Herein, we designed and synthesized four di-initiating organophosphorus superbases which, when combined with a bulky organoaluminum LA, enabled the efficient synthesis of well-defined tri-BCPs based on biorenewable MMBL at RT in a one-pot, two-step process (Fig. 1). The structural characterization of the key reaction intermediate and kinetic studies of the polymerization revealed that these bisphosphines indeed serve as covalently linked di-initiators and thus initiate the chain growth from both sides of the di-initiators concurrently. These BCPs appear to be the first example of MMBL-based tri-BCPs composed of both linear methacrylate and its cyclic analog (methylene butyrolactone) units by a living polymerization strategy. Overall, this LP polymerization system utilizing bisphosphine-based FLPs enabled the efficient, convenient synthesis of a series of tri-BCP TPEs containing high $T_g$ (227 °C) PMMBL as the two outer hard blocks and low $T_g$ (4 °C) poly(2-ethyoxyethyl methacrylate) (PEEMA) as the soft midblock. Tensile testing results revealed performance advantages of the biobased PMMBL TPEs over the petroleum-based PMMA TPEs, especially under high working temperature conditions.

## Results and discussion

**Synthesis of iminophosphine superbase di-initiators.** A straightforward method was developed for the synthesis of four bis(imino)phosphine superbases, with the same imidazoline framework [P(NI$^i$Pr)Ph] but varied in the length (spacing) of the alkyl bridging group -(CH$_2$)$_n$-: $n = 2$, $\mu^{Et}$; 3, $\mu^{Pr}$; 4, $\mu^{Bu}$; 6, $\mu^{Hex}$ (Fig. 1). Taking $\mu^{Bu}$[P(NI$^i$Pr)Ph]$_2$ as an example, commercially available bisphosphine precursor Ph$_2$PC$_4$H$_8$PPh$_2$ was treated with Li to generate the corresponding dilithium salt [PhPC$_4$H$_8$PPh]$^{2-}$Li$^+_2$(THF)$_4$, which was subsequently reacted with PCl$_3$ at −78 °C to furnish Ph(Cl)PC$_4$H$_8$P(Cl)Ph; the ensuing reaction of this dichloride intermediate with the deprotonation product of imine (NI$^i$Pr)H treated by $n$-butyllithium ($^n$BuLi) afforded the target product (Fig. 2a and Supplementary Information). NMR spectra confirmed the successful synthesis of all four bis(imino)phosphine superbases (Supplementary Figs. 1–27), and two of them, $\mu^{Bu}$[P(NI$^i$Pr)Ph]$_2$ and $\mu^{Hex}$[P(NI$^i$Pr)Ph]$_2$, have also been structurally verified by single-crystal X-ray diffraction (Fig. 2b, c).

**Characteristics of bisphosphine-based di-initiators in the model MMA polymerization.** To identify the most efficient LP system that can also prevent the backbiting chain-termination side reactions[48,52], we examined the synergistic effects of the combination of C$_6$-bridged di-initiator $\mu^{Hex}$[P(NI$^i$Pr)Ph]$_2$ with five LAs with different acidity and steric hindrance on MMA polymerization (Fig. 1). They possessed descending acidity as revealed by Gutmann−Beckett method (Al(C$_6$F$_5$)$_3$ (100%) > (BHT)$_2$AlMe (86%) ≈ (BHT)$_2$Al($^i$Bu) (85%) > (BHT)Al($^i$Bu)$_2$ (74%) > AlMe$_3$ (71%))[48,52]. It turned out that the LP composed of Al(C$_6$F$_5$)$_3$ with the highest acidity achieved the highest polymerization rate. However, this LP system gave a low initiating efficiency (run 1, Table 1) and unsuccessful chain extension (Supplementary runs 1 and 2, Table 3; Supplementary Fig. 37), presumably due to backbiting side reactions resulted from its high acidity that can effectively activate the in-chain ester carbonyl[48,52]. The use of AlMe$_3$ with the lowest acidity achieved incomplete MMA conversion up to 24 h (run 2, Table 1). With the stronger LA (BHT)Al$^i$Bu$_2$, the polymerization rate was

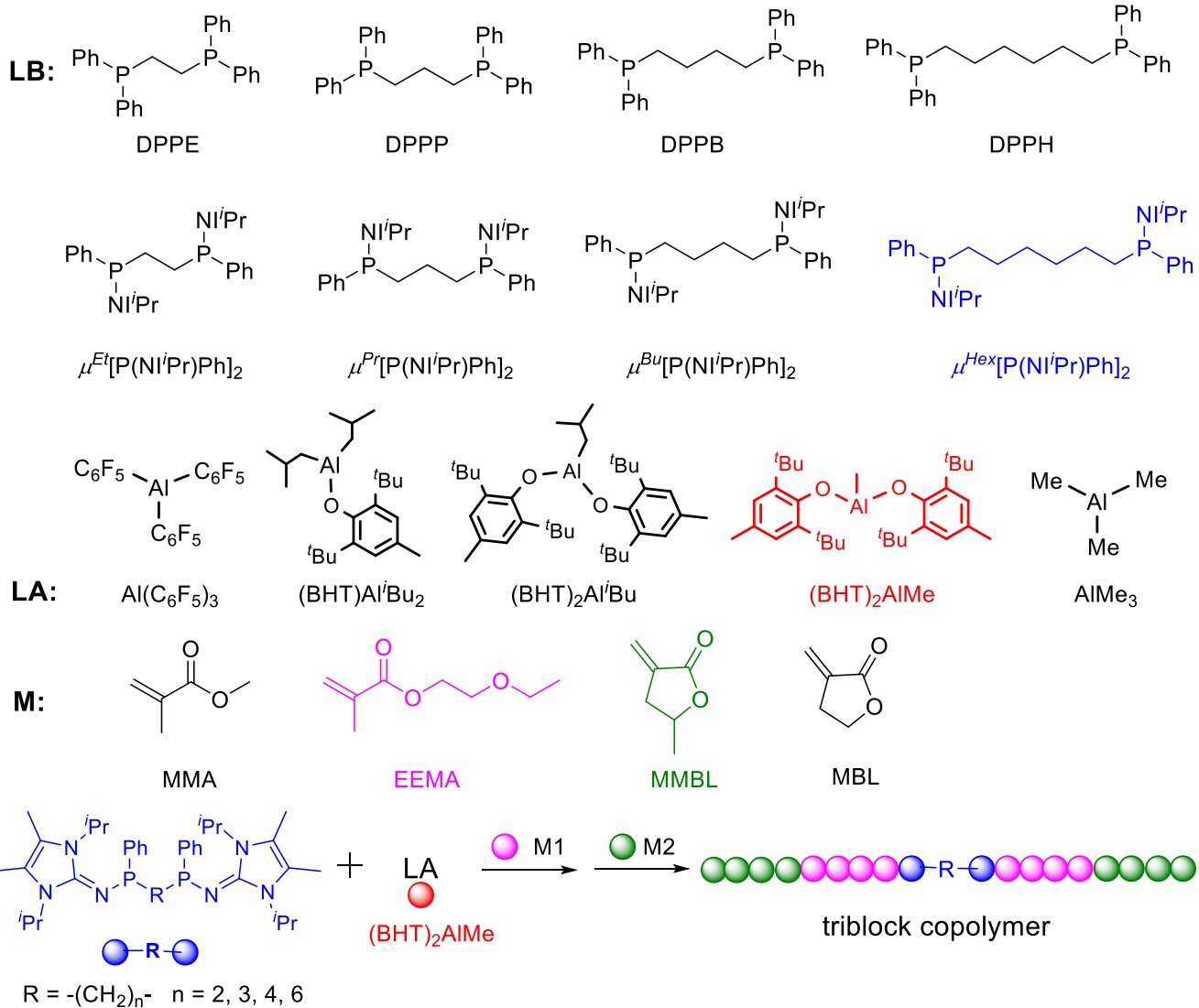

**Fig. 1 Structures and synthetic process.** The chemical structures of Lewis base (LB), Lewis acid (LA), and monomer (M) investigated in this study. Bottom: outlined synthesis of tri-BCPs in a one-pot, two-step process.

significantly enhanced relative to AlMe₃, but the resulting PMMA showed relatively broader dispersity (Đ = 1.37) and moderate initiation efficiency of 77% (run 3, Table 1). Switching to bulkier (BHT)₂Al$^i$Bu with further enhanced acidity led to a much enhanced initiation efficiency of 103%, but it took 540 min to reach full monomer consumption (run 4, Table 1). Finally, the use of (BHT)₂AlMe, a LA with comparable acidity but less steric hindrance relative to (BHT)₂Al$^i$Bu, rendered the rapid (full monomer conversion in 3 min) and controlled polymerization, producing PMMA having $M_n$ = 41.7 kg/mol (Đ = 1.20), essentially identical to that of the theoretical value, thus giving rise to a near quantitative $I^*$ of 97% (run 5, Table 1). Overall, in terms of polymerization characteristics, there is subtle interplay between the LA's acidity and steric hindrance as both aspects of the LA affect the degree of activation of monomer (good for enhancing the polymerization rate) and in-chain ester groups (bad for polymerization control) as well as the nucleophilicity and stabilization of the growing enolate chain end.

Accordingly, (BHT)₂AlMe was chosen as the paring LA for examining the effectiveness of the LPs when combined with each of the other three bisphosphine superbases. It turned out that all of the LPs could achieve quantitative monomer conversion within 10 min, but $\mu^{Et}$[P(NI$^i$Pr)Ph]₂ based exhibited the lowest initiation efficiency of 69% (runs 6–8, Table 1). Moreover, kinetic studies revealed that the length of the alkyl linker connecting both sides of the (imino)phophine imposes significant impact on the polymerization of MMA. Thus, increasing the length of the linker from -(CH₂)₂- to -(CH₂)₃-, -(CH₂)₄- and -(CH₂)₆- led to gradual enhancement of the polymerization rate, as indicated by the observed rate constants of 0.25/s to 0.60/s, 0.69/s and 0.91/s, respectively (with same concentrations of the LP in each run, Supplementary Fig. 28). Therefore, the LP (BHT)₂AlMe/$\mu^{Hex}$[P(NI$^i$Pr)Ph]₂ exhibited the best polymerization performance among the investigated LPs. For a comparative study, we also examined the effectiveness of four bisphosphine precursors with the same -(CH₂)ₙ- (n = 2, 3, 4, 6) linkers but without the imidazolin-2-ylidenamino (NI$^i$Pr) framework (Fig. 1), in combination with the same LA (BHT)₂AlMe on the polymerization of MMA. The PMMA products produced by these simple bisphosphines exhibited broader dispersity indices and the polymerization had low initiation efficiencies ($I^*$% = 9–16, runs 9–12, Table 1), highlighting the necessity of having the NI$^i$Pr framework in the bisphosphine structure for achieving the well-controlled MMA polymerization.

**Fig. 2 Synthesis and structures. a** synthetic route for bis(imino)phosphine superbase. **b** X-ray crystal structure of $\mu^{Bu}[P(NI^iPr)Ph]_2$. P(1)-N(1) 1.655(5) Å, C(9)-N(1) 1.334(6) Å, C(9)-N(2) 1.367(6) Å, C(9)-N(3) 1.388(5) Å. **c** X-ray crystal structure of $\mu^{Hex}[P(NI^iPr)Ph]_2$. P(1)-N(1) 1.6731(16) Å, C(4)-N(1) 1.300 (2) Å, C(4)-N(2) 1.385(6) Å, C(4)-N(3) 1.378(2) Å. Hydrogen atoms omitted for clarity and ellipsoids drawn at 50% probability.

**Table 1 Polymerization results by different LPs[a].**

| Run | LB | LA | M | [M]:[LA]:[LB] | Solvent | Time (min) | Conv.[b] (%) | $M_n$ ($M_w$)[c] (kg·mol⁻¹) | Đ[c] | I*[d] (%) |
|---|---|---|---|---|---|---|---|---|---|---|
| 1 | $\mu^{Hex}[P(NI^iPr)Ph]_2$ | Al(C₆F₅)₃ | MMA | 400:4:1 | TOL | 0.3 | 100 | 52.9 | 1.20 | 76 |
| 2 | $\mu^{Hex}[P(NI^iPr)Ph]_2$ | AlMe₃ | MMA | 400:4:1 | TOL | 1440 | 71.6 | n.d | n.d | n.d |
| 3 | $\mu^{Hex}[P(NI^iPr)Ph]_2$ | (BHT)Al$^i$Bu₂ | MMA | 400:4:1 | TOL | 0.5 | 100 | 52.2 | 1.37 | 77 |
| 4 | $\mu^{Hex}[P(NI^iPr)Ph]_2$ | (BHT)₂Al$^i$Bu | MMA | 400:4:1 | TOL | 540 | 100 | 39.6 | 1.28 | 103 |
| 5 | $\mu^{Hex}[P(NI^iPr)Ph]_2$ | (BHT)₂AlMe | MMA | 400:4:1 | TOL | 3 | 100 | 41.7 | 1.20 | 97 |
| 6 | $\mu^{Et}[P(NI^iPr)Ph]_2$ | (BHT)₂AlMe | MMA | 400:4:1 | TOL | 10 | 100 | 59.1 | 1.28 | 69 |
| 7 | $\mu^{Pr}[P(NI^iPr)Ph]_2$ | (BHT)₂AlMe | MMA | 400:4:1 | TOL | 4 | 100 | 39.2 | 1.22 | 103 |
| 8 | $\mu^{Bu}[P(NI^iPr)Ph]_2$ | (BHT)₂AlMe | MMA | 400:4:1 | TOL | 4 | 100 | 41.0 | 1.20 | 99 |
| 9 | DPPE | (BHT)₂AlMe | MMA | 400:4:1 | TOL | 30 | 100 | 434 | 1.49 | 9 |
| 10 | DPPP | (BHT)₂AlMe | MMA | 400:4:1 | TOL | 30 | 100 | 271 | 1.38 | 15 |
| 11 | DPPB | (BHT)₂AlMe | MMA | 400:4:1 | TOL | 20 | 100 | 255 | 1.36 | 16 |
| 12 | DPPH | (BHT)₂AlMe | MMA | 400:4:1 | TOL | 15 | 100 | 254 | 1.35 | 16 |
| 13[e] | $\mu^{Hex}[P(NI^iPr)Ph]_2$ | (BHT)₂AlMe | MMBL | 400:4:1 | DCM | 2 | 100 | (73.6) | 1.28 | 78 |
| 14[e] | $\mu^{Hex}[P(NI^iPr)Ph]_2$ | (BHT)₂AlMe | MMBL | 800:4:1 | DCM | 10 | 100 | (115) | 1.17 | 91 |
| 15[e] | $\mu^{Hex}[P(NI^iPr)Ph]_2$ | (BHT)₂AlMe | MMBL | 1600:4:1 | DCM | 120 | 100 | (200) | 1.07 | 96 |
| 16[e] | $\mu^{Hex}[P(NI^iPr)Ph]_2$ | (BHT)₂AlMe | MMBL | 3200:4:1 | DCM | 720 | 100 | (435) | 1.31 | 108 |

*TOL* toluene, *DCM* dichloromethane.
[a]Conditions: carried out at RT; [M]$_0$ = 0.936 mol/L.
[b]Monomer conversions measured by ¹H NMR.
[c]$M_n$ and Đ ($M_w/M_n$) determined by GPC relative to PMMA standards in DMF.
[d]Initiator efficiency (I*)% = $M_n$(calcd)/$M_n$(exptl) × 100, where $M_n$(calcd) = [MW(MMA)]([MMA]$_0$/[I]$_0$) (conversion) + MW of chain-end groups.
[e]Absolute molecular weight ($M_w$) measured by GPC coupled with a light scattering detector.

**Evidence for living polymerization of MMBL.** With such a well-controlled $\mu^{Hex}[P(NI^iPr)Ph]_2$/(BHT)₂AlMe FLP system in hand, we further interrogated the living features of the MMBL polymerization and obtained four lines of strong evidence to show that it is indeed a living polymerization system. First, for polymerization performed with [MMBL]$_0$/[LP]$_0$ ratios varied from 400/1, 800/1, 1600/1 to 3200/1, the (BHT)₂AlMe/$\mu^{Hex}[P(NI^iPr)Ph]_2$ FLP system can rapidly and quantitatively convert MMBL into PMMBL with medium to high molecular weights, narrow molecular weight distributions and high to near quantitative initiation efficiencies (runs 13–16, Table 1, Fig. 3a). Second, a plot of the PMMBL $M_n$ value and Đ vs monomer conversion at a fixed [MMBL]/[$\mu^{Hex}[P(NI^iPr)Ph]_2$] ratio of 800/1 gave a straight line

($R^2 = 0.998$), which was coupled with low Đ values in the range of 1.10–1.12 (Supplementary Fig. 36). Third, successful chain extension experiments provide more direct evidence for the living nature of the $\mu^{Hex}[P(NI^iPr)Ph]_2$/(BHT)₂AlMe LP system (Supplementary runs 1 and 2, Table 1). Fourth, as the bis(imino) phosphine was employed as a di-initiator, it is expected to initiate two polymer chains propagating from both sides of the central bisphosphine. Therefore, upon full consumption of the first pool of monomers, the sequential addition of the second pool of different monomers should enable the further propagation of the polymer chain from the both sides, thus furnishing a tri-BCP. Through this one-pot, two-step process, the well-defined tri-BCP PMBL-*b*-PMMBL-*b*-PMBL was prepared by using MBL as co-

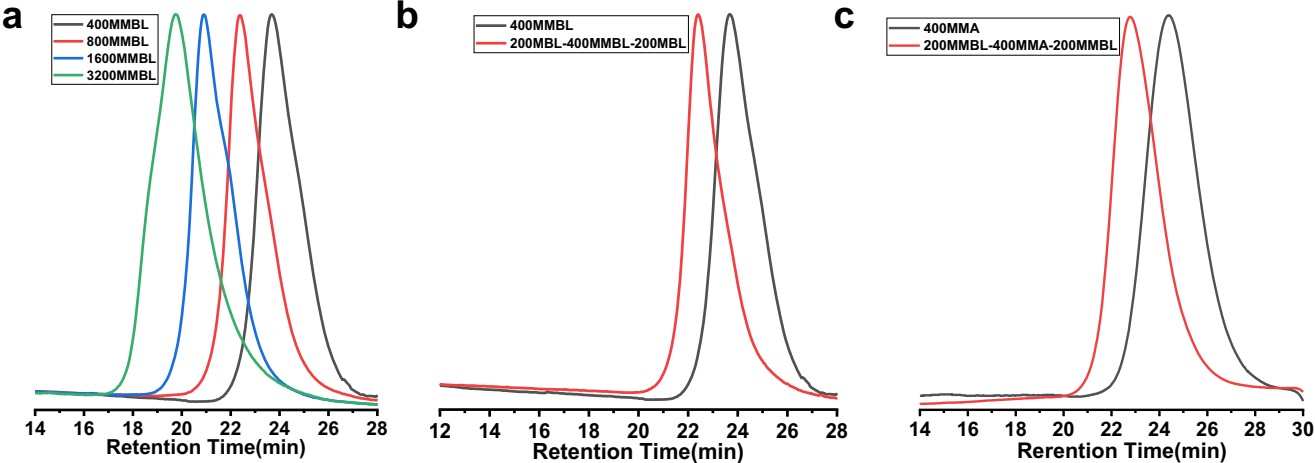

**Fig. 3 GPC analysis. a** GPC traces of PMMBL produced by $\mu^{Hex}[P(NI^{i}Pr)Ph]_2/(BHT)_2AlMe$ LP at various $[MMBL]_0/[\mu^{Hex}[P(NI^{i}Pr)Ph]_2]_0/[(BHT)_2AlMe]_0$ ratio at RT. Conditions: $[MMBL]_0/[\mu^{Hex}[P(NI^{i}Pr)Ph]_2]_0/[(BHT)_2AlMe]_0 = 400{:}1{:}4, 800{:}1{:}4, 1600{:}1{:}4, 3200{:}1{:}4$, $[MMBL]_0 = 0.936$ M. **b** GPC traces of homopolymer and tri-BCP produced from the sequential block copolymerization of MMBL and MBL by $\mu^{Hex}[P(NI^{i}Pr)Ph]_2/(BHT)_2AlMe$ in dichloromethane at RT, $[MMBL]_0 = 0.936$ M. **c** GPC traces of homopolymer and tri-BCP produced from the sequential block copolymerization of MMA and MMBL by $\mu^{Hex}[P(NI^{i}Pr)Ph]_2/(BHT)_2AlMe$ in toluene at RT, $[MMA]_0 = 0.936$ M.

monomer (Fig. 3b, Supplementary, run 3, Table 1). More significantly, by utilizing MMBL as the co-monomer for polymerization of MMA, we also successfully synthesized tri-BCP PMMBL-*b*-PMMA-*b*-PMMBL (Fig. 3c, Supplementary run 4, Table 1). To the best of our knowledge, it represents the first example of a tri-BCPs composed of both cyclic and linear polar acrylic monomers by the living polymerization strategy to date.

**Mechanistic aspects of polymerization**. To gain more insights into the polymerization mechanism, we performed a series of in-situ NMR reactions at RT. A noninteracting, true FLP was generated upon mixing $\mu^{Hex}[P(NI^{i}Pr)Ph]_2$ with $(BHT)_2AlMe$ (Supplementary Figs. 29 and 30), which is essential for achieving both the high polymerization rate and degree of polymerization control[52, 60]. Furthermore, the stoichiometric reaction of $\mu^{Hex}[P(NI^{i}Pr)Ph]_2$ with $(BHT)_2AlMe{\cdot}MMA$ in a 1:2 molar ratio yielded the corresponding bis-zwitterionic enolaluminate intermediate $(BHT)_2MeAl\text{-}O(MeO)C{=}(Me)CCH_2\text{-}\mu^{Hex}[P(NI^{i}Pr)Ph]_2\text{-}CH_2C(Me){=}C(OMe)O\text{-}Al(BHT)_2Me$ (Supplementary Figs. 33 and 34). The solid-state structure was confirmed by the single-crystal analysis of its analogs $(BHT)_2MeAl\text{-}O(MeO)C{=}(Me)CCH_2\text{-}\mu^{Bu}[P(NI^{i}Pr)Ph]_2\text{-}CH_2C(Me){=}C(OMe)O\text{-}Al(BHT)_2Me$, by replacing $\mu^{Hex}[P(NI^{i}Pr)Ph]_2$ with $\mu^{Bu}[P(NI^{i}Pr)Ph]_2$ (Fig. 4, Supplementary Figs. 31 and 32). The single-crystal structure clearly shows that each (imino)phosphine base site of the di-initiator underwent nucleophilic attack at the LA-activated monomer leading to the generation of the corresponding bis-zwitterionic active species. Compared with the bisphosphine di-initiator $\mu^{Bu}[P(NI^{i}Pr)Ph]_2$ before the reaction, the distance for P(1)–N(1) in the bis-zwitterionic enolaluminate intermediate (biszwitterion **1**) is shortened from 1.655(5) Å to 1.568(3) Å, whereas the C(1)-N(1) bond length did not show obvious changes. This important crystal structural information enabled us to better understand this LP polymerization system and confirmed the dual initiation of the designed bis(imino)phosphine di-initiators. Therefore, these results established strong basis for the accelerated synthesis of tri-BCPs through a one-pot, two-step sequential addition of monomer method. After the isolation and purification of biszwitterion **1**, we employed it to perform a control polymerization experiment in a 400:1 [MMA]/[biszwitterion **1**] ratio and found that the biszwitterion alone is ineffective for MMA polymerization, and no monomer conversion was detected up to 16 h (Supplementary

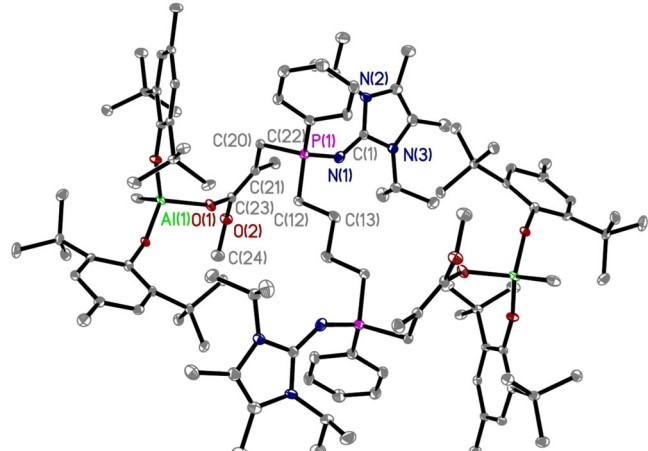

**Fig. 4 Structural characterization.** X-ray crystal structure of $(BHT)_2MeAl\text{-}O(MeO)C{=}(Me)CCH_2\text{-}\mu^{Bu}[P(NI^{i}Pr)Ph]_2\text{-}CH_2C(Me){=}C(OMe)O\text{-}Al(BHT)_2Me$ (P(1)-N(1) 1.568(3) Å, C(1)-N(1) 1.352(5) Å, C(1)-N(2) 1.373(4) Å, C(1)-N(3) 1.357(5) Å). Hydrogen atoms omitted for clarity and ellipsoids drawn at 30% probability.

run 3, Table 3). However, upon addition of another two equiv. of $(BHT)_2AlMe$ to the above mixture, full monomer consumption was achieved in only 4 min, furnishing PMMA with $M_n = 34.3$ kg/mol ($\mathit{Đ} = 1.17$) (Supplementary run 4, Table 3). These results indicated that the activation of monomer by LAs is essential to achieve an effective LPP system.

Next, we examined the chain-propagation kinetics of polymerization of MMA with varied $[MMA]/[\mu^{Hex}[P(NI^{i}Pr)Ph]_2]/[(BHT)_2AlMe]$ ratios of 1600:1:3, 1600:1:4, 1600:1:5, and 1600:1:6. The representative kinetic plots of $[MMA]_t/[MMA]_0$ vs time clearly revealed no induction time and a strict zero-order dependence on [MMA] concentration for all the ratios (Fig. 5a). A double-logarithm plot (insert) of the apparent rate constants ($K_{app}$), obtained from the slopes of the best-fit lines to the plots of $[MMA]_t/[MMA]_0$ vs time, as a function of $\ln[(BHT)_2AlMe]$ was fit to a straight line ($R^2 = 0.9937$) with a slope of ~1.0, revealing that the propagation is first order in $[(BHT)_2AlMe]$ concentration. In the second set of kinetic experiments, with a fixed amount

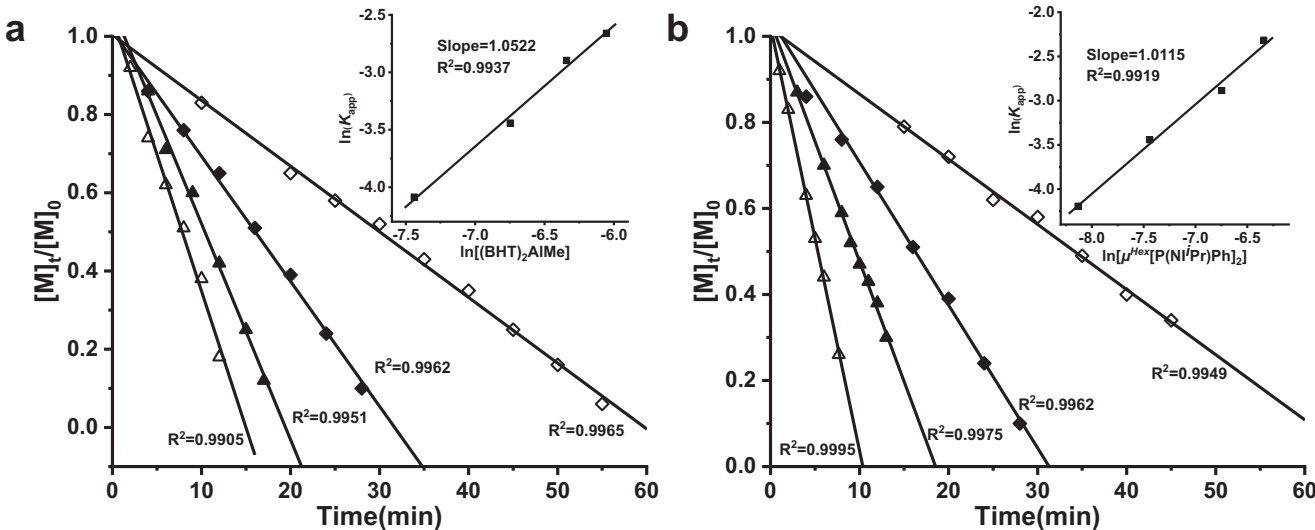

**Fig. 5 Kinetic studies. a** Zero-order kinetic plots for polymerization of MMA by $(BHT)_2AlMe/\mu^{Hex}[P(NI^iPr)Ph]_2$ in toluene at RT: $[MMA]_0 = 0.936$ M; $[\mu^{Hex}[P(NI^iPr)Ph]_2]_0 = 0.588$ mM; $[(BHT)_2AlMe]_0 = 1.764$ mM ($\diamondsuit$), 2.352 mM ($\blacklozenge$), 2.94 mM ($\blacktriangle$), 3.528 mM ($\triangle$). Inset: plot of $\ln(k_{app})$ vs ln $[(BHT)_2AlMe]$. **b** Zero-order kinetic plots for polymerization of MMA by $(BHT)_2AlMe/\mu^{Hex}[P(NI^iPr)Ph]_2$ in toluene at RT: $[MMA]_0 = 0.936$ M; $[(BHT)_2AlMe]_0 = 1.175$ mM; $[\mu^{Hex}[P(NI^iPr)Ph]_2]_0 = 0.294$ mM ($\diamondsuit$), 0.588 mM ($\blacklozenge$), 1.175 mM ($\blacktriangle$), 1.764 mM ($\triangle$). Inset: plot of $\ln(k_{app})$ vs ln $[\mu^{Hex}[P(NI^iPr)Ph]_2]$.

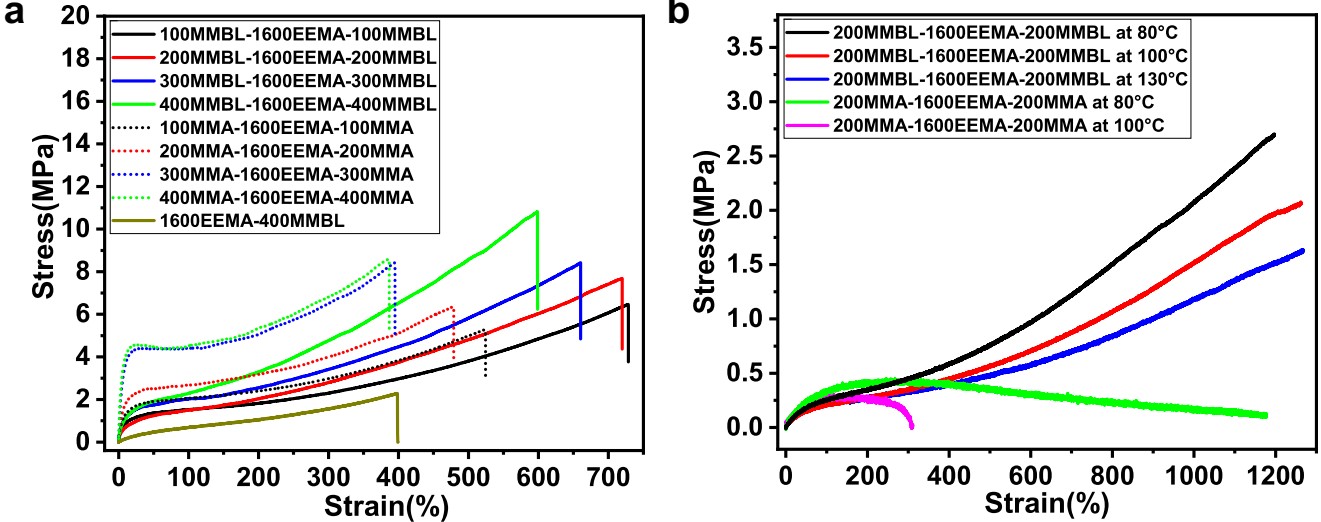

**Fig. 6 Mechanism.** Proposed mechanism for polymerization via dual initiation by $\mu^{Hex}[P(NI^iPr)Ph]_2$-based LP.

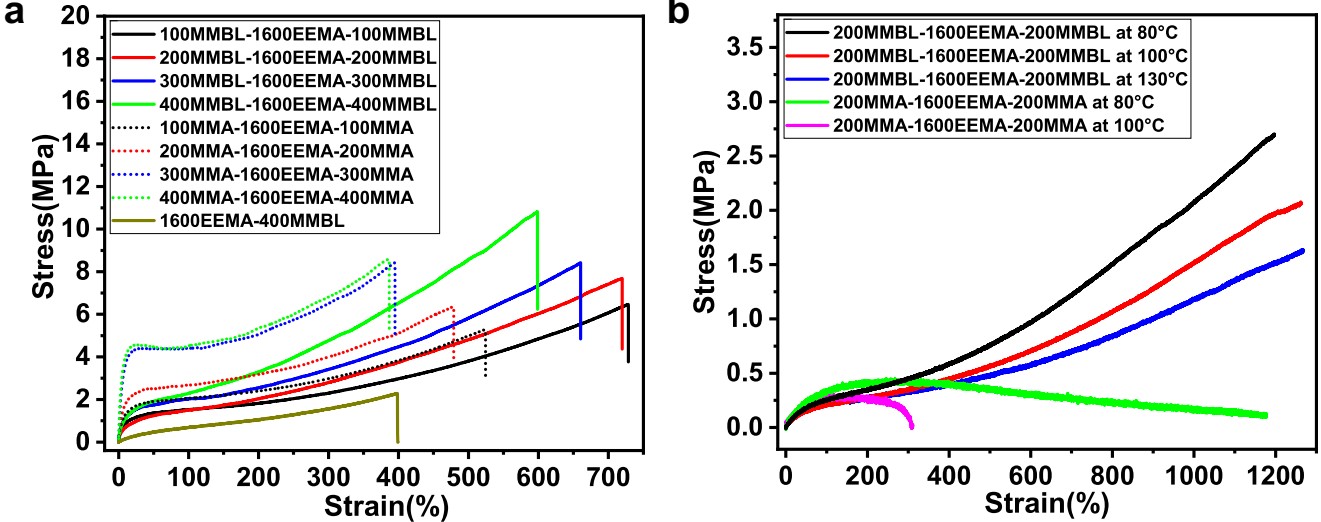

**Fig. 7 Mechanical performance. a.** Comparison of the stress−strain curves of different ratios of PMMBL-based TPEs with those of PMMA-based TPEs at RT. **b** Comparison of the stress−strain curves of PMMBL-based TPEs with those of PMMA-based TPEs at high temperatures (80–130 °C).

of $[MMA]_0$, the $[(BHT)_2AlMe]_0/[\mu^{Hex}[P(NI^iPr)Ph]_2]_0$ ratio was varied at 8:3, 6:2, 4:1, and 3:0.5 such that at each ratio there were 2 equiv. of $(BHT)_2AlMe$ left to activate the monomer upon formation of the active intermediate that consumes $\mu^{Hex}[P(NI^iPr)Ph]_2$ and $(BHT)_2AlMe$ in 1:2 ratio. The same zero-order dependence on monomer concentration was observed for all of the ratios investigated in this study (Fig. 5b). A double-logarithm

plot (insert) of the apparent rate constants ($K_{app}$), obtained from the slopes of the best-fit lines to the plots of $[MMA]_t/[MMA]_0$ vs time, as a function of $\ln[\mu^{Hex}[P(NI^iPr)Ph]_2]$ was fit to a straight line ($R^2 = 0.9919$) with a slope of ~1.0, revealing that the propagation is first order in $[\mu^{Hex}[P(NI^iPr)Ph]_2]$ concentration. These kinetics results are consistent with what we have observed for the LP polymerization with monoinitiators[48]. Since the

employed $\mu^{Hex}$[P(NI$^i$Pr)Ph]$_2$ has two P sites, each phosphine group individually initiated one polymer chain growth starting from the central bisphosphine. Based on the structure of the key reaction intermediate, as well as the results of the polymerization and kinetic studies, we proposed a mechanism for the polymerization of MMA by $\mu^{Hex}$[P(NI$^i$Pr)Ph]$_2$/(BHT)$_2$AlMe FLP shown in Fig. 6.

In the proposed mechanism, the bisphosphine di-initiator undergoes nucleophilic attack (with the two Lewis basic P sites) onto two individual molecules of monomer activated by (BHT)$_2$AlMe from both sides, generating the corresponding bis-zwitterionic active species. The ensuing intermolecular Michael addition of the active species to the incoming LA-activated monomer leads to the propagation of the polymer chains from both ends of the central bisphosphine initiator (Fig. 6). Overall, this mechanism is consistent with the results obtained to date and also provides fundamental basis for the synthesis of well-defined tri-BCPs from our one-pot, two-step sequential monomer additional method enabled by the current robust bis(imino)phosphine superbase di-initiator.

**PMMBL-based tri-BCPs as high-temperature TPEs.** As described above, PMMBL, due to its high $T_g$ and bio-renewability, should be an ideal hard segment candidate for bio-based TPE synthesis. To this end and by taking advantage of accelerated tri-BCP synthesis uniquely enabled by the current bis(imino)phosphine superbase di-initiators, we have synthesized a series of ABA type tri-BCP TPEs (Supplementary Table 2) by using PMMBL as the hard end blocks and PEEMA as the soft midblock ($T_g = 4$ °C)[53]. For comparison, we also synthesized tri-BCP TPEs using PMMA as the hard end blocks, instead of PMMBL. To measure their mechanical properties, colorless, transparent, air-bubble-free TPEs films of the tri-BCPs were prepared by the solvent-casting method (Supplementary Fig. 35). Uniaxial tensile testing of the PMMBL-based TPEs at RT showed that when the soft segment PEEMA fixed at 1600, increasing the hard segment contents from 200 to 400, 600, 800, the tensile strength increased from 6.46 to 7.67, 8.42 and 10.8 MPa, whereas the elongation at break gradually decreased from 728% to 719%, 660% and 598%, respectively (Fig. 7a, Supplementary Table 2). As for TPEs using PMMA as hard segments, smaller elongation at break and lower tensile strength were observed for TPEs with hard segments of 200 or 400. However, yield points were clearly observed when the contents of hard segments were further increased to 600 or 800 (Fig. 7a). We also synthesized di-block copolymer 1600EEMA-400MMBL and found that it exhibited inferior mechanic properties to that of tri-BCPs (Fig. 7a), thus highlighting the importance of tri-BCPs in the application as TPEs.

These RT tensile testing results indicated that the mechanical properties of the MMBL-based TPEs are superior to those of PMMA-based tri-BCPs. Their mechanical property differences are much more pronounced at higher temperatures. For example, the PMMA-based tri-BCP is no longer a TPE at 80 °C (just a weak elastomer) and rapidly breaks at 100 °C (Fig. 7b). In sharp contrast, the PMMBL-based BCPs still maintained TPE properties upon heating at 80 °C, 100 °C and even 130 °C. Although the tensile strength becomes lower relative to that measured at RT, the corresponding elongation at break is much higher (>1200%, Fig. 7b). Collectively, the above results demonstrate considerably superior mechanical properties of the PMMBL-based tri-BCPs to those of PMMA-based tri-BCPs, especially at high temperature, suggesting potential for the PMMBL tri-BCP as high-temperature TPEs.

In summary, we have designed and synthesized four novel bis(imino)phosphine superbases featuring the alkyl linker of the series with different spacing between the two P sites. When combined with bulky organoaluminum LAs, they form non-interacting FLPs, which promote rapid and living polymerization of MMBL, furnishing polymers with predictable molecular weights, low dispersity indices, and high to near quantitative initiation efficiencies. The length of bridging alkyl group is found to have a marked impact on the polymerization activity and the degree of polymerization control. On balance, $\mu^{Hex}$[P(NI$^i$Pr)Ph]$_2$/(BHT)$_2$AlMe FLP with the longest alkyl linker exhibited the best polymerization performance and living features among the investigated FLPs. Mechanistic studies, including the single crystal structure analysis of the bis-zwitterionic active species, as well as detailed polymerization and kinetic studies, enabled the elucidation of the polymerization mechanism; this mechanism features the unique dual initiation and subsequent chain growth from both P sites of the bis(imino)phosphine superbase di-initiator. Thanks to such unique features of our designed di-initiators, we accomplished the efficient and accelerated synthesis of both bio-derived PMMBL and petroleum-derived PMMA-based tri-BCPs from a one-pot, two-step procedure. Significantly, compared with PMMA-based tri-BCP TPEs, the PMMBL-based tri-BCP TPEs exhibited considerably superior mechanical properties, especially at high working temperatures, and maintained TPE properties even at 130 °C, suggesting their potential as high-temperature TPEs.

## Methods

**Synthesis of N, N-(ethane-1,2-diylbis(phenylphosphanediyl))bis(1,3-diisopropyl-4,5-dimethyl-1,3- dihydro-2H-imidazol-2-imine) ($\mu^{Et}$[P(NI$^i$Pr)Ph]$_2$).** It was prepared in the similar procedure as described for the synthesis of P(NI$^i$Pr)Ph$_2$[52]. In an argon-filled glovebox, a 200 mL Schlenk flask was equipped with a stir bar and charged with THF (80 mL) and (NI$^i$Pr)H (1.95 g, 10 mmol). This flask was sealed with a rubber septum, removed from the glovebox, interfaced to a Schlenk line, and then brought to −78 °C, where a solution of n-BuLi (1.6 M in hexane, 6.25 mL, 10 mmol) was added dropwise via syringe to the above flask. After completion of the addition, the cold bath was removed and the mixture was allowed to warm to RT and stirred at RT for 3 h. It was cooled down to −78 °C again and added with Ph(Cl)PC$_2$H$_4$P(Cl)Ph[61] (1.58 g, 5 mmol, in 20 mL THF), then warmed to RT and stirred overnight. The volatiles were removed in vacuo and the residue (50:50 mix of rac and meso diastereomers) was dissolved in 90 mL of hexane. After filtration, the filtrate was collected and concentrated to 20 mL, then stored in the freezer, $\mu^{Et}$[P(NI$^i$Pr)Ph]$_2$ was obtained as white solid (about 1:8 mix of rac and meso diastereomers). Yield 45% (1.43 g, 2.26 mmol).

See Supplementary Information for detailed methods and characterization data related to the synthesis of bifunctional organophosphorus superbases.

**General polymerization procedures.** Polymerizations were performed in 20 mL glass reactors inside the glovebox for ambient temperature (ca. 25 °C) runs. In a typical polymerization procedure, a predetermined amount of a Lewis acid (LA) (4 equiv.), such as (BHT)Al$^i$Bu$_2$·MMA, was first dissolved in 500 μL of MMA and toluene inside a glovebox. The polymerization was started by rapid addition of $\mu^{Et}$[P(NI$^i$Pr)Ph]$_2$ solution (1 equiv.) via a gastight syringe to the above mixture under vigorous stirring. After the measured time interval, a 0.2 mL aliquot was taken from the reaction mixture via pipet and quickly quenched into a 4-mL vial containing 0.6 mL of undried "wet" CDCl$_3$ stabilized by 250 ppm of BHT-H; the quenched aliquots were later analyzed by $^1$H NMR to obtain the percent monomer conversion data. After the polymerization was stirred for the stated reaction time then the reactor was taken out of the glovebox, and the reaction was quenched by addition of 5 mL of 5% HCl-acidified methanol. The quenched mixture was isolated by filtration and dried in a vacuum oven at 50 °C to a constant weight.

## Data availability

All the characterization data and experimental protocols are provided in this article and its Supplementary Information. The X-ray crystallographic data for compounds $\mu^{Bu}$[P(NI$^i$Pr)Ph]$_2$, $\mu^{Hex}$[P(NI$^i$Pr)Ph]$_2$ and (BHT)$_2$MeAl-O(MeO)C=(Me)CCH$_2$-$\mu^{Bu}$[P(NI$^i$Pr)Ph]$_2$-CH$_2$C(Me)= C(OMe)O-Al(BHT)$_2$Me have been deposited at the Cambridge Crystallographic Data Center (CCDC), under deposition number 2064107, 2064108 and 2064109. These data can be obtained free of charge from CCDC via www.ccdc.cam.ac.uk/data_request/cif.

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

## Acknowledgements

This work was supported by the National Natural Science Foundation of China (Grant nos. 22071077, 21774042, 21871107 and 21975102) to Y.Z. and the U. S. National Foundation (NSF-1904962) to E.Y.X.C.

## Author contributions

Y.B. designed and conducted experiments. H.W. performed the experiments and characterizations. Y.B., J.H. and Y.Z. analyzed the data and wrote the manuscript with input from all other authors. E.Y.X.C, discussed the results and edited the manuscript.

## Competing interests

The authors declare no competing interests.
