## [Peer Review File · Nature Communications]

Reviewers' Comments:

Reviewer #1:

Remarks to the Author:

The manuscript reports the synthesis of high-molecular-weight ABA triblock copolymers of a renewable cyclic methacrylate, gamma-methyl-alpha-methylene-gamma-butyrolactone (MMBL), and 2-ethoxyethyl methacrylate (EEMA) using a series of newly synthesized bis(imino)phosphine Lewis superbases in the presence of a bulky Lewis acid. I would like to highly appreciate successful results on the synthesis of bis(imino)phosphine bases and bifunctional initiation from the compound for formation of ABA triblock copolymers with high molecular weight and good mechanical properties. However, the corresponding author has already reported the synthesis of high-molecular-weight PMMBL using similar frustrated Lewis pairs although the initiator is monofunctional (reference 57). In addition, similar acrylic bio-based ABA thermoplastic elastomers have already been reported using a similar monomer, alpha-methylene-gamma-butyrolactone (MBL), but different systems, by several researchers (e.g., *Macromolecules* 41, 5509-5511, 2008; *Polymer* 50, 2087-2094, 2009; *Biomacromolecules* 13, 3833-3840, 2012). Although the molecular weights and mechanical properties reported in this article are superior to those reported, the concepts of renewable ABA acrylic bio-based thermoplastic elastomers are the same. So, I would not recommend publication in *Nature Communications*. It is more suitable for publication in a more specific polymer journal.

Reviewer #2:

Remarks to the Author:

In their contribution, Zhang and co-workers describe the Lewis Pair Polymerization (LPP) of several acrylic monomers, including MMA and its cyclic analogue MMBL. The authors succeed in the controlled preparation of ABA-triblock copolymers by application of a one-pot, two-step procedure based on a difunctional initiator (bis(imino)phosphines) in combination with tailored Al-type Lewis acids. That way, it was possible to create block copolymers that behave as thermoplastic elastomers, employing either PMMA- or PMMBL-blocks as the "hard" terminal segments.

The paper is well written and all data clearly presented. Conclusions are well supported by experimental evidence. Albeit LPP is now becoming a polymerization methodology which is increasingly employed – and the polymerization system presented here shows all the typical characteristics of LPP, as convincingly demonstrated by the authors – the work described in this manuscript is a very fine example of LPP and addresses an interesting aspect (mechanical properties of renewably sourced materials). I thus think this contribution represents a significant advance in this field to justify publication in *Nature Communications*.

Some minor points might be considered beforehand:

- In Figure S21 (31P-NMR) there seem to be some additional signals. Can these be assigned?
- It is intriguing to note that $k(\text{App})$ seems to be impacted so heavily by the linker length. Is there a (preliminary) explanation for this behavior?
- I think it would be helpful to the reader if the properties of the Al-type LA and their impact on the polymerization mechanism would be discussed in some more detail. For example, the fact that the lowest acidity (BHT)Al(iBu)₂ engenders the fastest polymerization rate seems to indicate that in this case monomer activation is less important than interaction with the propagating enolate.
- In the final section (i.e., Figure 5) it might be helpful to show (or at least mention) the stress-strain performance of PMMA/PMMBL homopolymer and the corresponding diblock copolymers (i.e., PEEMA-PMMBL) as a context for the triblock copolymers investigated here.

Reviewer #3:

Remarks to the Author:

This is an interesting and clever use of FLPs in polymerization. The PI has previously demonstrated the utility of FLPs in the initiation of polymerization, however here he cleverly modifies the system so as to incorporate bis-phosphine backbones into the polymer. The nature of the bis-phosphine has a dramatic impact on the polymer properties. While I cannot comment on the property/materials characterization of polymers as this is outside my expertise, I can say that the use of the new bis(imino)phosphine superbases in FLP chemistry are unprecedented. In this regard, I did wonder if rather than increasing the basicity of the phosphine, the Lewis acidity of the Al species could be increased and give a similar result?

Other than that the work appears well, thoroughly characterized and certainly interesting to both the inorganic and polymer communities.

Changes Made in the Revision and Responses to the Comments by the Reviewers

Reviewer: 1

Reviewer's general comments: The manuscript reports the synthesis of high-molecular-weight ABA triblock copolymers of a renewable cyclic methacrylate, gamma-methyl-alpha-methylene-gamma-butyrolactone (MMBL), and 2-ethoxyethyl methacrylate (EEMA) using a series of newly synthesized bis(imino)phosphine Lewis superbases in the presence of a bulky Lewis acid. I would like to highly appreciate successful results on the synthesis of bis(imino)phosphine bases and bifunctional initiation from the compound for formation of ABA triblock copolymers with high molecular weight and good mechanical properties. However, the corresponding author has already reported the synthesis of high-molecular-weight PMMBL using similar frustrated Lewis pairs although the initiator is monofunctional (reference 57). In addition, similar acrylic bio-based ABA thermoplastic elastomers have already been reported using a similar monomer, alpha-methylene-gamma-butyrolactone (MBL), but different systems, by several researchers (e.g., *Macromolecules* 41, 5509-5511, 2008; *Polymer* 50, 2087-2094, 2009; *Biomacromolecules* 13, 3833-3840, 2012). Although the molecular weights and mechanical properties reported in this article are superior to those reported, the concepts of renewable ABA acrylic bio-based thermoplastic elastomers are the same. So, I would not recommend publication in *Nature Communications*. It is more suitable for publication in a more specific polymer journal.

Our response: We appreciated your valuable time spent reviewing this manuscript and your critical comments. While we agree with your general comments on the context of this work, we disagree with your characterization of this work for being lack of conceptual novelty in view of the published work as we firmly believe that this manuscript bears sufficient novelty and originality compared with the previous work specifically mentioned in your review.

First of all, we would like to clarify that the high molecular PMMBL mentioned in your review was actually produced by a *non-living* FLP catalyst system based on commercially available phosphines or N-heterocyclic carbenes (ref. 57: *Angew. Chem. Int. Ed.* 2010, **49**, 10158-10162). Those FLPs could not precisely control the molecular weight and molecular weight distribution of the resulting polymers. As far as we know, none of the reported monofunctional FLP catalyst system achieved the synthesis of bio-based cyclic acrylic ABA-type triblock copolymers due to their drastically different reactivities compared with the linear alkyl methacrylate monomers, which present challenges in crossover propagation. In this manuscript, we specifically designed and synthesized bis(imino)phosphine Lewis superbases to solve this challenge. Although synthetically challenging, especially concerning the isolation and structural characterization of the unprecedented biszwitterionic active species, we endeavored to successfully synthesize a series of bis(imino)phosphine superbases and zwitterionic active species, as well as structurally characterize them by NMR spectroscopy and single-crystal X-ray crystallography. Owing to the unique dual initiation and propagation of polymer chains from both P sites of the propagating species as well as the living/controlled features of such dinuclear FLP systems, for the first time, we were able to achieve the bio-acrylic ABA-type triblock copolymers in a simple one-pot, two-step process. Therefore, the motivation and strategy described in this manuscript depart significantly from the previous systems, either from a perspective of the FLP design, synthesis and characterization, or from a perspective of the polymer synthesis and properties.

Second, we appreciated your listing of the three relevant references cited for similarity and thus the argument for the lack of conceptual novelty in the design and synthesis of renewable acrylic ABA thermoplastic elastomers. We agree that the polymer composition is similar (not the same), but the employed polymerization strategies or methods are drastically different between the current system and those already reported, and the current dual-initiating and living FLP system is much superior in several aspects. For the atom transfer radical polymerization

(ATRP) method utilized in the first two cited publications (Macromolecules, 2008; Polymer, 2009), a macroinitiator (Br-PBA-Br) needs to be synthesized and isolated at no more than 53% monomer conversion before addition of MBL for chain extension to make the triblock copolymer. For the third paper (Biomacromolecules 2012), a macroinitiator was also required for the synthesis of block copolymers, which was prepared by ring-opening transesterification polymerization of menthide, followed by transformation into α,ω -dibromo end-functionalized poly(menthide) (Br-PM-Br). In short, the above reported strategies require isolation and purification of the macroinitiator, involving multiple synthetic steps. In sharp contrast, our living/controlled FLP strategy can achieve the ABA-type triblock copolymers with predicted molecular weight and narrow molecular weight distribution simply through a one-pot, two sequential addition process. Furthermore, it took up to 22 hours for ATRP to obtain incomplete monomer conversion whereas quantitative monomer conversion was reached within several minutes by our FLP strategy. Therefore, the current FLP strategy produces triblock copolymers much more rapidly, conveniently, and efficiently, enabling the synthesis of high molecular weight triblock copolymers exhibiting superior mechanical properties, as compared to the ATRP method. In our revised manuscript, we analyzed these similarities and key differences.

In summary, we argue that the current dinuclear dual-initiating and living FLP system is unique not only from the perspective of the new FLP design, synthesis, and characterization, but also from the perspective of its unprecedented efficiency and livingness in polymerization of highly reactive, bio-based cyclic acyclic monomers and crossover propagation between linear and cyclic acrylic monomers for the synthesis of well-defined bio-based ABA triblock copolymers as high-temperature TPEs. Thus, we firmly believed that this manuscript has sufficient novelty and originality compared with the previous work.

Reviewer: 2

Reviewer's general comments: In their contribution, Zhang and co-workers describe the Lewis Pair Polymerization (LPP) of several acrylic monomers, including MMA and its cyclic analogue MMBL. The authors succeed in the controlled preparation of ABA-triblock copolymers by application of a one-pot, two-step procedure based on a difunctional initiator (bis(imino)phosphines) in combination with tailored Al-type Lewis acids. That way, it was possible to create block copolymers that behave as thermoplastic elastomers, employing either PMMA- or PMMBL-blocks as the “hard” terminal segments.

The paper is well written and all data cleanly presented. Conclusions are well supported by experimental evidence. Albeit LPP is now becoming a polymerization methodology which is increasingly employed – and the polymerization system presented here shows all the typical characteristics of LPP, as convincingly demonstrated by the authors – the work described in this manuscript is a very fine example of LPP and addresses an interesting aspect (mechanical properties of renewably sourced materials). I thus think this contribution represents a significant advance in this field to justify publication in Nature Communications.

Our response: We greatly appreciated your comments and recommendation for publication in *Nature Communications* after minor revisions. We carefully considered your following specific comments and made corresponding changes in the revised manuscript.

Reviewer's general comments:

(1) In Figure S21 (31P-NMR) there seem to be some additional signals. Can these be assigned?

Our response: The peak centered around -25ppm should be attributed to the unreacted 1,2-bis(diphenylphosphino)hexane, which is now noted in the Figure caption.

(2) It is intriguing to note that $k(\text{App})$ seems to be impacted so heavily by the linker length. Is there a (preliminary) explanation for this behavior?

Our response: According to the proposed polymerization mechanism, both phosphorus sites of difunctional initiator (bis(imino)phosphines) would simultaneously initiate polymerization through the nucleophilic attack of the activated monomer. Therefore, the steric hindrance around the phosphorus sites should significantly affect the rate of the polymerization, especially chain initiation. With increasing the linker length, the steric hindrance around the phosphorus centers is decreased, resulting in an increase in the polymerization activity accordingly.

(3) I think it would be helpful to the reader if the properties of the Al-type LA and their impact on the polymerization mechanism would be discussed in some more detail. For example, the fact that the lowest acidity (BHT)Al(iBu)₂ engenders the fastest polymerization rate seems to indicate that in this case monomer activation is less important than interaction with the propagating enolate.

Our response: We appreciated the valuable suggestion! During the polymerization, the monomer was firstly activated by the organoaluminum LA, followed by nucleophilic attack by a bis(imino)phosphine base to generate the corresponding biszwitterion, which is the active species for propagation of the polymer chain. To proceed the polymerization, an extra amount of the LA is needed to activate the monomer. After isolation and purification of the biszwitterion, we employed it directly for MMA polymerization as a control experiment in a 400:1 [MMA]/[biszwitterion] ratio. As expected, we found that biszwitterion is ineffective for MMA polymerization in the absence of another equiv. of the LA (the catalyst to activate the monomer), and no monomer conversion was detected in up to 16 h. Upon addition of (BHT)₂AlMe to the above mixture, it took only 5 min to reach full monomer consumption. These results indicated that the activation of monomer by the LAs is essential to achieve an effective LPP system.

Among the three investigated organoaluminum LAs employed in the original version of the manuscript, (BHT)Al(iBu)₂ with the relatively lower acidity exhibited high polymerization activity, presumably due to its smaller steric hindrance compared with the other two LAs and less stabilization of the enolate (thus stronger nucleophile), although it provides a lower degree of monomer activation. According to the reviewer's suggestion, we also included two more LAs (Al(C₆F₅)₃ with higher acidity and AlMe₃ with lower acidity) to systematically investigate the influence of acidity and steric hindrance of the LAs on polymerization. The relevant experimental results and discussion were added to the revised manuscript.

(4) In the final section (i.e., Figure 5) it might be helpful to show (or at least mention) the stress-strain performance of PMMA/PMMBL homopolymer and the corresponding diblock copolymers (i.e., PEEMA-PMMBL) as a context for the triblock copolymers investigated here.

Our response: Uniaxial tensile testing (as shown below) indicated that the PMMA film produced by hot pressing exhibited a high tensile strength of 50 MPa but an extremely low elongation at break of 3%, suggesting the brittleness of PMMA (which is consistent with literature reports). PMMBL is too brittle to prepare a suitable film for tensile testing no matter what hot-pressing or solvent casting method is employed. The tensile testing also indicated that the diblock 1600PEEMA-*b*-400PMMBL copolymers produced by the FLP system based on the monofunctional phosphorus superbase only exhibited a low tensile strength around 1 MPa and an elongation at break no more than 400%. These relevant experimental data and description have been included into the revised manuscript.

Stress/strain curve of PMMA obtained by uniaxial tensile testing.

Reviewer: 3

Reviewer's general comments: This is an interesting and clever use of FLPs in polymerization. The PI has previously demonstrated the utility of FLPs in the initiation of polymerization, however here he cleverly modifies the system so as to incorporate bis-phosphine backbones into the polymer. The nature of the bis-phosphine has a dramatic impact on the polymer properties. While I cannot comment on the property/materials characterization of polymers as this is outside my expertise, I can say that the use of the new bis(imino)phosphine superbases in FLP chemistry are unprecedented. In this regard, I did wonder if rather than increasing the basicity of the phosphine, the Lewis acidity of the Al species could be increased and give a similar result? Other than that the work appears well, thoroughly characterized and certainly interesting to both the inorganic and polymer communities.

Our response: We greatly appreciated the reviewer's comments! As suggested, we added two more organoaluminum LAs ($\text{Al}(\text{C}_6\text{F}_5)_3$ with higher acidity and AlMe_3 with lower acidity) to systematically investigate the influence of acidity and steric hindrance of the LAs on polymerization. The five LAs investigated in this study possessed descending acidity as revealed by Gutmann–Beckett method: $\text{Al}(\text{C}_6\text{F}_5)_3$ (100%) > $(\text{BHT})_2\text{AlMe}$ (86%) \approx $(\text{BHT})_2\text{Al}^i\text{Bu}$ (85%) > $(\text{BHT})\text{Al}^i\text{Bu}_2$ (74%) > AlMe_3 (71%). It turned out that the LP composed of $\text{Al}(\text{C}_6\text{F}_5)_3$ with the highest acidity achieved the highest polymerization rate. However, this LP system gave a low initiating efficiency and unsuccessful chain extension, most likely due to back-biting side reactions resulted from its high acidity (activating the in-chain ester carbonyl). The use of AlMe_3 with the lowest acidity in the series achieved incomplete MMA conversion in up to 24 h. With the stronger acid $(\text{BHT})\text{Al}^i\text{Bu}_2$, polymerization rate was significantly enhanced relative to AlMe_3 , but the resulting PMMA showed relatively broader dispersity (1.37) and lower initiation efficiency (77%). Switching to bulkier $(\text{BHT})_2\text{Al}^i\text{Bu}$ with further enhanced acidity led to near quantitative initiation efficiency, but drastically decreased the polymerization rate. Finally, the employment of $(\text{BHT})_2\text{AlMe}$, a LA with similar acidity but less steric hindrance relative to $(\text{BHT})_2\text{Al}^i\text{Bu}$ rapidly and quantitatively polymerized MMA into PMMA with predicted molecular weight and narrow molecular weight distribution, furnishing near quantitative initiation efficiency. Therefore, $(\text{BHT})_2\text{AlMe}$ was selected to examine its effectiveness on the polymerization when combined with bisphosphines. In short, in terms of polymerization characteristics, there is subtle interplay between the LA's acidity and steric hindrance as both aspects of the LA affect the degree of activation of monomer (good for enhancing the polymerization rate) and in-chain ester groups (bad for polymerization control) as well as the nucleophilicity and stabilization of the growing enolate chain end. The relevant polymerization data and description have been included into the revised manuscript.

Reviewers' Comments:

Reviewer #1:

Remarks to the Author:

Thank you for your detailed replies for my comments. I know and understand the point of this paper. In my opinion or in the authors' reply, I understand that the most important point of this paper is the synthesis of bis(imino)phosphine Lewis superbases, isolation and characterization of unprecedented biszwitterionic active species, and the development of novel excellent living polymerizations with dual active sites. I have already highly appreciated this achievement in my previous comments. However, the concept of the renewable acrylic ABA biobased polymers is not new and similar polymers have already been prepared. Although the performance is better than the reported, the differences or the high performance are difficult to notice. In addition, I am not sure how important this material is. In my opinion, the bio-based thermoplastic polymer is just one of the many applications of this excellent polymerization but is not the main dish. Although bio-based materials are hot and may sound more attractive, the main part of this paper may be different. If this paper is accepted for publication, I would recommend that the title may be changed and focuses on the novel living polymerization.

Reviewer #2:

Remarks to the Author:

In their revised manuscript, Zhang and co-workers have included additional information and experiments on mechanical properties (homo- and diblock copolymers) and on a wider range of Al-based Lewis acids.

All concerns raised in my previous report have been addressed sufficiently. I think this manuscript is a very fine example for Lewis-Pair-Polymerization, making best use of the controlled characteristics of this method to create material with improved properties. I recommend publication of this work in its current form.

Reviewer #3:

Remarks to the Author:

I have reviewed the author's response, the referee's comments as well as the revised manuscript. I believe the authors have addressed the concerns regarding the range of Lewis acidity in an appropriate manner.

Again, while I cannot provide expertise on the property/materials characterization of polymers, my impression is the authors have responded professionally and thoroughly. Thus I am supportive of publication in Nature Commun.

Changes Made in the Revision and Responses to the Comments by the Editor and Reviewers

Addressing the Editor's comments and instructions on formatting:

1. A revised author checklist describing your response to our editorial requests

Answer: Provided as requested.

2. A separate point-by-point response to the reviewers' comments, reproduced verbatim.

Answer: A point-by-point response letter to the comments of reviewer has been provided.

3. The final version of your manuscript as a Word file, with all changes highlighted in the text and any tables prepared using the table menu in Word.

Answer: The manuscript highlighted with all changes has been provided in word file.

3. Production-quality versions of each figure as a separate file containing all panels.

Answer: Figures in high quality were provided in a PPT file as instructed.

4. Any updated checklists that verify compliance with our research ethics and data reporting standards in PDF format.

Answer: provided as instructed.

5. The final version of the Supplementary Information in one PDF file

Answer: provided as instructed.

Reviewer: 1

Reviewer's general comments: Thank you for your detailed replies for my comments. I know and understand the point of this paper. In my opinion or in the authors' reply, I understand that the most important point of this paper is the synthesis of bis(imino)phosphine Lewis superbases, isolation and characterization of unprecedented biszwitterionic active species, and the development of novel excellent living polymerizations with dual active sites. I have already highly appreciated this achievement in my previous comments. However, the concept of the renewable acrylic ABA biobased polymers is not new and similar polymers have already been prepared. Although the performance is better than the reported, the differences or the high performance are difficult to notice. In addition, I am not sure how important this material is. In my opinion, the bio-based thermoplastic polymer is just one of the many applications of this excellent polymerization but is not the main dish. Although bio-based materials are hot and may sound more attractive, the main part of this paper may be different. If this paper is accepted for publication, I would recommend that the title may be changed and focuses on the novel living polymerization.

Our response: We greatly appreciated your valuable suggestion and change the title as "Dual-Initiating and Living Frustrated Lewis Pairs: Expeditious Synthesis of Biobased Thermoplastic Elastomers" as suggested.

Reviewer: 2

Reviewer's general comments: In their revised manuscript, Zhang and co-workers have included additional information and experiments on mechanical properties (homo- and diblock

copolymers) and on a wider range of Al-based Lewis acids.

All concerns raised in my previous report have been addressed sufficiently. I think this manuscript is a very fine example for Lewis-Pair-Polymerization, making best use of the controlled characteristics of this method to create material with improved properties. I recommend publication of this work in its current form.

Our response: We greatly appreciated your kind recommendation for publication in *Nature Communications* in its current form.

Reviewer: 3

Reviewer's general comments: I have reviewed the author's response, the referee's comments as well as the revised manuscript. I believe the authors have addressed the concerns regarding the range of Lewis acidity in an appropriate manner.

Again, while I cannot provide expertise on the property/materials characterization of polymers, my impression is the authors have responded professionally and thoroughly.

Thus I am supportive of publication in Nature Commun.

Our response: We greatly appreciated your kind recommendation for publication in *Nature Communications*.